# Paraspinal Muscle Fat Infiltration as a Key Predictor of Symptomatic Intravertebral Vacuum Cleft: A Machine Learning Approach

**DOI:** 10.3390/jcm14093109

**Published:** 2025-04-30

**Authors:** Joonghyun Ahn, Jaewan Soh, Young-Hoon Kim, Jae Chul Lee, Jun-Seok Lee, Hyung-Youl Park, Jeong-Han Lee, June Lee, Youjin Shin

**Affiliations:** 1Department of Orthopedic Surgery, Bucheon St. Mary’s Hospital, College of Medicine, The Catholic University of Korea, Bucheon 14647, Republic of Korea; ajhssnim@gmail.com (J.A.);; 2Department of Orthopaedic Surgery, Hanyang University Guri Hospital, College of Medicine, Hanyang University, Guri 11923, Republic of Korea; md962014@gmail.com; 3Department of Orthopedic Surgery, Seoul St. Mary’s Hospital, College of Medicine, The Catholic University of Korea, Seoul 06591, Republic of Korea; boscoa@catholic.ac.kr; 4Department of Orthopaedic Surgery, Soonchunhyang University Hospital, College of Medicine, Soonchunhyang University, Seoul 04401, Republic of Korea; jlee@schmc.ac.kr; 5Department of Orthopedic Surgery, Eunpyeong St. Mary’s Hospital, College of Medicine, The Catholic University of Korea, Seoul 03312, Republic of Korea; junband@naver.com (J.-S.L.); matrixbest@naver.com (H.-Y.P.); 6Department of Data Science, The Catholic University of Korea, Bucheon 14662, Republic of Korea; leejune0502@catholic.ac.kr

**Keywords:** machine learning, intravertebral vacuum cleft, vertebral compression fracture, risk factor

## Abstract

**Background/Objectives**: Symptomatic intravertebral vacuum cleft (SIVC) is a complication of vertebral compression fractures (VCFs) that leads to persistent pain and deformity. Its prediction remains challenging due to multifactorial causes. Paraspinal muscle fat infiltration has been associated with spinal fracture outcomes but has not been extensively explored in SIVC prediction. Our aim was to develop machine learning (ML) models for predicting SIVC and to evaluate the role of muscle-related variables in improving predictive performance. **Methods**: Demographic, radiological, and muscle-related variables were collected. ML models—including Logistic Regression, Random Forest, XGBoost, and Multi-Layer Perceptron—were trained and tested under two input conditions: baseline variables (SETTING_1) and baseline plus muscle-related variables (SETTING_2). Model performance was evaluated using accuracy, the area under the receiver operating characteristic curve (AUC), and feature importance analysis. **Results**: The Random Forest model in SETTING_2, which incorporated muscle-related variables, achieved the highest accuracy (96.6%) and AUC (0.956). Multifidus fatty infiltration (MFfi), erector spinae fatty infiltration (ESfi), and endplate CSA were identified as the most significant predictors. The inclusion of muscle-related variables significantly improved the predictive performance of all ML models. **Conclusions**: ML models, particularly Random Forest, demonstrated high accuracy in predicting SIVC when muscle-related variables were included. Paraspinal muscle fat infiltration is a critical predictor of SIVC and should be integrated into risk assessment strategies to improve early diagnosis and management.

## 1. Introduction

The vertebral compression fracture (VCF) is a common spinal condition that has been increasing in prevalence in recent decades. If unresolved through conservative treatment due to various factors, it can lead to persistent back pain, kyphotic deformity, and neurological deterioration [1,2,3,4,5]. A hallmark of this condition is the intravertebral vacuum cleft (IVC), a distinctive radiological feature associated with the non-union of the VCF, some of which requires surgical intervention [6,7]. “Symptomatic” IVC (SIVC) represents a distinct clinical entity where patients with VCFs develop recurrent pain after initial improvement, accompanied by characteristic radiographic findings of IVC.

Unlike typical VCF cases that show spontaneous improvement, SIVC patients experience persistent or recurring pain (VAS > 3) that correlates with specific radiological features. The early detection and accurate prediction of SIVC are clinically significant, yet these remain challenging due to the multifactorial nature of its pathogenesis [8,9]. SIVC has been reported to occur slightly more frequently in females, with other risk factors including renal disease, diabetes, chronic steroid use, osteoporosis, alcoholism, hypothyroidism, and radiation therapy [6,8,10,11,12,13,14]. Recent studies have also reported that fat infiltration of the paraspinal muscles may increase the risk of VCFs or influence clinical and radiological outcomes following spinal fractures [15,16,17]. Specifically, fatty infiltration in the multifidus (MF) and erector spinae (ES) muscles has been shown to reduce spinal support, increasing the risk of osteoporotic vertebral compression fractures and potentially leading to higher rates of non-union following fusion surgery [17]. The quantitative assessment of muscle status could serve as an important biomarker for early diagnosis and treatment planning for SIVC, which have direct clinical implications for patient management and outcome improvement.

In recent decades, the rapid development of machine learning (ML) algorithms has significantly impacted orthopedic surgery. ML enables the high-accuracy analysis and prediction of complex patterns in medical imaging, electronic medical records (EMRs), and other clinical datasets [18]. Previous studies have demonstrated the utility of ML in predicting osteoporotic VCFs using magnetic resonance imaging (MRI) and demographic data alongside bone mineral density scores [19,20]. However, to our knowledge, no previous studies have employed ML approaches incorporating muscle-related variables to predict VCFs and SIVC.

Therefore, the primary objective of this study was to determine whether muscle-related variables were significant risk factors in the prediction of SIVC using ML by comparing demographics and variables measured from plain radiography with and without muscle-related variables. The secondary objective was to investigate which ML algorithms could best predict SIVC and to what extent muscle-related variables had significant effects.

## 2. Materials and Methods

### 2.1. Study Design and Patient Population

This study was a retrospective analysis of patients diagnosed with VCFs who were enrolled between March 2013 and February 2023 (Figure 1). This study was approved by the Institutional Review Board (IRB) of the institution. The inclusion criteria were as follows: patients diagnosed with a VCF at the thoracolumbar junction (TLJ and T10-L2) and patients diagnosed with SIVC at the TLJ. The exclusion criteria were patients with a pathological VCF, such as tumors or infection, patients with multiple-location VCFs, patients without MRI, and patients without adequate plain thoracolumbar anteroposterior and lateral radiographs.

### 2.2. Definition and Diagnostic Criteria of SIVC

SIVC was defined by the following criteria:

Clinical Criteria:
-Initial acute pain following a VCF.-Period of initial improvement with conservative treatment.-Development of recurrent pain (VAS > 3).-Location-specific pain corresponding to the level of vacuum cleft.Radiographic Criteria:
-Radiographic evidence of intravertebral vacuum cleft (IVC) on plain thoracolumbar radiographs.-Absence of other significant pathology at the affected and adjacent levels, such as new fractures.

To address potential diagnostic subjectivity, two independent spine specialists (YHK and JA) confirmed the diagnosis of SIVC.

### 2.3. Data Collection

A clinical data warehouse (CDW) containing all the medical records from the institution was used to select and study patients. Demographic and radiological data were collected from all enrolled patients. The demographic data included sex, age, diabetes, hypertension, adrenal insufficiency, hyperthyroidism, hypothyroidism, and steroid use.

Radiological characteristics were measured using the following variables: the “angle” (local kyphotic angle) and compression ratio of the VCF were measured from the lateral view of the plane radiograph (PR) (Figure 2).

From the axial view of MRI, using ImageJ software (version 1.8.0, National Institutes of Health, Bethesda, MD (Maryland), USA), the cross-sectional areas (CSAs) of the VCF upper endplate, multifidus (MF), and erector spinae (ES) muscles were measured. Additionally, the percentage of fatty infiltration in MF (MFfi) and ES (ESfi) was quantified. For measuring fat infiltration, T2-weighted axial MRI images at the level of the vertebral fracture upper endplate were analyzed.

Regions of interest (ROIs) were manually drawn to outline the boundaries of the multifidus (MF) and erector spinae (ES) muscles bilaterally. The software’s threshold function was then applied to differentiate muscle tissue from fat based on signal intensity. The percentage of fatty infiltration was calculated as follows: (area of fat/total muscle area) × 100 (Figure 3). The relative multifidus (rMF) was defined as the MF/CSA of the endplate, and the relative erector spinae (rES) was defined as the ES/CSA of the endplate. All measurements were independently conducted by two trained observers (JA and YHK) who were blinded to the patient’s clinical information.

To evaluate inter-observer reliability, intraclass correlation coefficients (ICCs) were calculated for both MFfi and ESfi measurements on a random subset of 50 patients. The ICC for MFfi was 0.92 (95% CI: 0.89–0.95) and for ESfi was 0.89 (95% CI: 0.85–0.93), indicating excellent inter-observer reliability. For cases with measurement discrepancies exceeding 10%, a consensus was reached through joint reassessment. The final values used in the analysis were the average of the two observers’ measurements.

To address the class imbalance in our training dataset (654 VCF patients without IVC vs. 40 patients with SIVC), we applied the Synthetic Minority Over-Sampling Technique (SMOTE). Specifically, the original training set contained 28 SIVC cases (approximately 5.8%), which was increased to 457 cases after the application of SMOTE to achieve balance with the 458 VCF patients. This allowed the models to better capture patterns associated with the minority class while maintaining generalizability.

### 2.4. Machine Learning Models

To confirm the known risk factors for VCFs and SIVC, we conducted experiments using variables extracted from PR and EMR for SETTING_1 and variables extracted from PR, EMR, and MRI for SETTING_2. The patients were randomly divided into training (70%) and test (30%) datasets. A training set was used to develop the model. Supplementary 5-fold cross-validation was used to optimize the hyperparameters of the machine learning model in the training set, and the data of the training set were augmented using the SMOTE package to effectively learn from fewer datasets. The test set was used to evaluate the performance of the learned models: Logistic Regression (LR) [21], Random Forest (RF) [22], extreme gradient boosting (XGBoost) [23], and Multi-Layer Perceptron (MLP) [24]. Training and testing processes were conducted in Python 3.9.12 and PyCharm environments. 

### 2.5. Evaluation Matrix

For objective performance evaluation, the areas under the accuracy, specificity, sensitivity, precision, F1-score, and area under the receiver operating characteristics curve (AUROC) of the models were measured and averaged ten times for each method in the test set.

## 3. Results

### 3.1. Patient Characteristics

Of the 1694 patients, data from 694 patients who met the inclusion criteria and did not violate the exclusion criteria were included in this study; of these, 485 were used for training the ML models and 209 were used for testing. The analyses were summarized between SIVC and non-SIVC groups across multiple variables in Table 1. Notably, muscle-related variables (MFfi, ESfi, rMF, rES) and radiological measurements (angle, compression value, disc CSA) showed significant differences with *p* < 0.05.

### 3.2. Performances of Machine Learning Models

The LR, RF, XGBoost, and MLP models were evaluated on both the training and the test sets under SETTING_1 and SETTING_2.

In the training set of SETTING_1, the AUROC values were 0.911, 0.913, 0.853, and 0.863, respectively, with corresponding accuracies of 0.923, 0.940, 0.935, and 0.891 (Table 2). In the test set of SETTING_1, the AUROC values dropped to 0.699, 0.698, 0.643, and 0.708, and the accuracies were 0.757, 0.871, 0.856, and 0.760, respectively (Figure 4, Table 3).

In SETTING_2, all models demonstrated superior performance compared to SETTING_1. In the training set, the AUROCs were 0.963, 0.973, 0.967, and 0.923, and the accuracies were 0.973, 0.990, 0.993, and 0.972, respectively (Table 2). In the test set, the AUROCs were 0.947, 0.956, 0.951, and 0.904, and the accuracies were 0.951, 0.966, 0.962, and 0.961, respectively (Figure 5, Table 3).

All ML models showed strong generalizability from training to test datasets in SETTING_2. Among them, the RF model achieved the highest accuracy and AUROC on the training set, while the MLP model maintained the best test set AUROC performance, demonstrating its robustness across the settings. To evaluate model stability, we analyzed performance metrics across each fold of the 5-fold cross-validation process. Table 4 shows the AUC and accuracy of the Random Forest model in SETTING_2 (including muscle variables) for each fold.

### 3.3. Feature Importance

Figure 6 illustrates the feature importance ranking of the RF model in SETTING_1, where ‘Angle’ (local kyphotic angle), age, and ‘Solondo’ (steroid use) were identified as the top three predictors. In contrast, the RF model in SETTING_2, which outperformed all other models (Figure 7), highlighted MFfi, ESfi, and ‘Disc’ (endplate CSA) as the most influential predictors for SIVC, with feature importance values of 0.871, 0.702, and 0.575, respectively. Notably, feature importance values range from 0 to 1, indicating their relative contribution to the model’s predictive performance.

### 3.4. LIME Analysis

To enhance the interpretability of our model’s predictions, we applied the LIME algorithm to visualize the local decision boundaries of the Random Forest classifier for a representative SIVC case. Beyond identifying global feature importance, we conducted LIME analysis to gain insights into how specific combinations of features contribute to individual predictions and to examine the potential interrelationships between them in a localized context. As shown in Figure 8, the model relied most heavily on paraspinal muscle fat infiltration indicators, particularly MF% (>33.89%) and ES% (>25.96%), to predict SIVC. Additionally, other features such as low values of rMF and rES also positively contributed to the SIVC prediction. Conversely, the absence of hyperthyroidism and the presence of hypertension were among the features that marginally suppressed the prediction toward the SIVC class. These findings further support the global feature importance trends and suggest that muscle quality and vertebral biomechanical changes play a critical role in model prediction (Figure 8).

## 4. Discussion

In this study, we successfully predicted VCFs and SIVC using muscle-related variables using ML models. The superior performance of SETTING_2 (incorporating muscle variables) over SETTING_1 (conventional variables only) suggests that muscle-related factors play a crucial role in SIVC development. The high feature importance of MFfi and ESfi (0.871 and 0.702, respectively) indicates that paravertebral muscle status may be more predictive of SIVC than traditional risk factors such as age and steroid use. This will also provide insights into the accurate diagnosis of SIVC in the future. To the best of our knowledge, this is the first study to explore SIVC using machine learning (ML) in conjunction with muscle-related variables. To date, SIVC has primarily been identified through radiological assessments. However, challenges in accurately distinguishing SIVC from VCFs can result in inadequate treatment. Given that SIVC requires distinct treatments, such as bone cement augmentation, and has unique clinical features and prognoses compared with VCFs, precise identification is essential [25].

The exploratory analysis in Table 1 revealed significant differences between SIVC and non-SIVC groups across multiple muscle-related variables (MFfi, ESfi, rMF, rES), and radiological measurements (angle, compression value, disc CSA) showed significant differences with *p* < 0.05. Based on this initial analysis, we could formulate the following hypotheses: (1) The degree of fatty infiltration in multifidus and erector spinae muscles (MFfi and ESfi) likely represents strong predictive factors for SIVC occurrence (Figure 9). (2) Changes in the vertebral endplate cross-sectional area (disc CSA) are associated with SIVC development. (3) While age appears to be a significant predictor, muscle-related variables may have more direct associations with SIVC.

In line with our hypothesis, both MFfi and ESfi were ranked among the top three predictors. Previous studies have reported that MF and ES play crucial roles in maintaining spinal balance [26,27]. This suggests that fat infiltration weakens the supportive capacity of the spine, potentially contributing to the development of SIVC. Consistent with this, previous studies have reported that fat infiltration in the ES and MF is associated with reduced spinal bone mineral density (BMD) [28,29], while fat infiltration in the MF has been linked to an increased risk of osteoporotic VCFs [15]. In addition, it is noteworthy that high fat infiltration is associated with a low union rate following lumbar interbody fusion procedures [30,31]. This suggests that fat infiltration in the posterior muscles significantly affects the stability of the spinal column. In this study, the role of fat infiltration in the paravertebral muscles in the occurrence of SIVC was confirmed using feature importance analysis.

The enlargement of the endplate cross-sectional area (CSA), potentially resulting from the formation of intravertebral clefts, reflects biomechanical changes within the vertebral structure. These changes may represent a compensatory response to redistribute the mechanical load or indicate progressive instability associated with advanced spinal degeneration [32]. This finding emphasizes the complex nature of endplate pathology and its important role in SIVC development. Furthermore, structural changes in the endplate CSA could serve as valuable biomarkers for identifying patients at risk of SIVC [33]. However, further research is required to establish whether this predictor has a causal role or simply correlates with the condition and to determine how it can be integrated into diagnostic or therapeutic approaches.

The use of steroids has been identified as a contributing risk factor for the development of SIVC [6]. In this study, SETTING_2 demonstrated lower feature importance values for steroid use compared to other factors, while SETTING_1 highlighted its significance as one of the top three predictors. However, high-quality studies specifically investigating the relationship between steroid use and SIVC are limited. Notably, the long-term improper use of steroids has been reported to increase the risk of new vertebral fractures, suggesting that inappropriate steroid use compromises vertebral stability [34,35].

Diabetes has been reported as a risk factor for SIVC and has been reported to be strongly associated with the frailty index, which implies a significant impact on overall health and resilience [12]. Several mechanisms have been demonstrated to explain this relationship. Diabetes often contributes to muscle loss (or sarcopenia) and weakness, both essential components of frailty [36]. Nutritional challenges associated with diabetes, including altered dietary patterns and nutrient absorption, can further intensify frailty [37]. Additionally, diabetes is commonly accompanied by comorbidities such as cardiovascular disease, obesity, and kidney disease, all of which significantly affect the frailty burden [38]. While some studies suggest that diabetes does not always have a direct association with SIVC, its long-term complications and systemic effects contribute to its potential role in diminishing physical and structural resilience [39,40]. This, in turn, could indirectly impact outcomes such as vertebral stability and the risk of SIVC.

Age has been reported as a significant risk factor for the occurrence of past VCFs [41,42]. In this study, the age parameter showed significant results in SETTING_1, but the age parameter in SETTING_2 did not show significant results compared to other variables. This means that muscle-related variables representing an individual’s health status are more important than chronological age in the fracture-healing process. Of course, age should not be overlooked in the occurrence of SIVC because there have been reports that age, osteoporosis, and sarcopenia are correlated [43]. However, it can be seen that it should be interpreted differently based on the meaning of risk factors mentioned in previous studies [36].

This study provides data-driven insights and valuable medical perspectives. With the development of computer and data science, grafting into the medical field is taking place. This provides an accurate identification and diagnosis of SIVC and can lead to the discovery of risk factors through data analysis.

Nevertheless, several limitations should be considered when interpreting our results. First, the retrospective nature of this study meant that standardized pain scores were not available throughout the follow-up period, potentially affecting our ability to fully characterize pain progression patterns. Second, while we carefully defined SIVC criteria, the distinction between SIVC-related pain and other sources of chronic pain relies partly on clinical judgment. Third, potential contributing factors such as the frailty index [44], BMI, and BMD were excluded due to missing data. Fourth, our focus on single-level fractures, while improving study homogeneity, limits generalizability to multilevel cases. Finally, the relatively small number of SIVC cases (n = 40) suggests the need for larger validation studies. Given these limitations, future studies should investigate the potential clinical application of muscle-related predictors in the early diagnosis and treatment of SIVC, further validating their role in improving patient outcomes.

## 5. Conclusions

This study identified paravertebral muscle variables as novel risk factors for SIVC and highlighted effective prediction methods. From a clinical perspective, the fatty infiltration of multifidus and erector spinae muscles represents a key biomarker for predicting SIVC occurrence, suggesting that the careful evaluation of muscle status should be integrated into routine MRI assessment. Machine learning models, particularly Random Forest, can effectively integrate complex clinical and radiological data to predict SIVC with high accuracy, offering potential as clinical decision support tools.

## Figures and Tables

**Figure 1 jcm-14-03109-f001:**
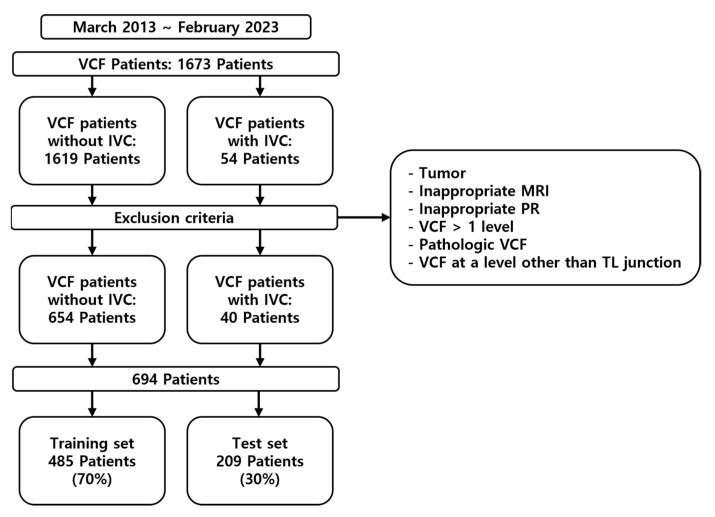
Diagram illustrating the data collection process and the segmentation process into training and test sets.

**Figure 2 jcm-14-03109-f002:**
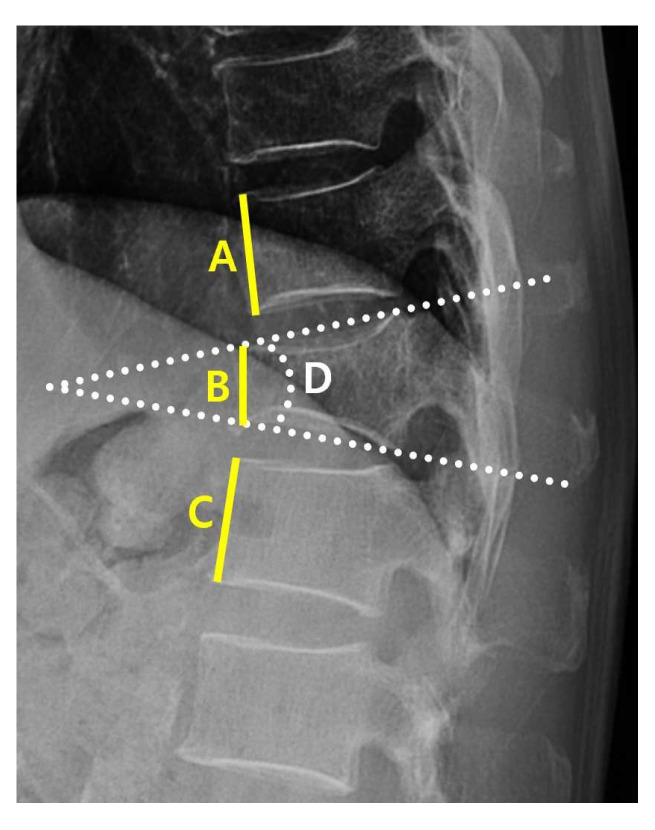
Measurement of kyphotic angle and compression value. The kyphotic angle and compression value are measured within the picture archiving and communication (PACS) system PR lateral view; D shows the kyphotic angle of the T11 fracture; the compression value is calculated as ((A + C)/2) − B)/((A + C)/2).

**Figure 3 jcm-14-03109-f003:**
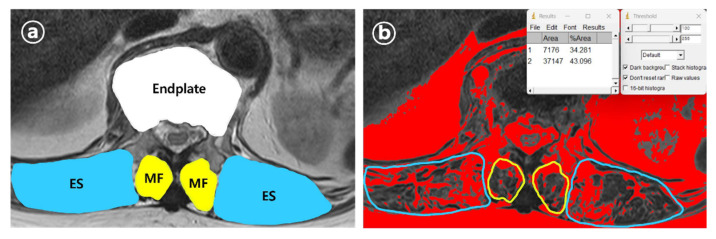
Measurement variables from axial view. (**a**) Measurement of CSA of endplate, MF, and ES of VCF location, measured within the PACS system MRI axial view. The CSA of MF and ES was calculated as the mean of the left and right CSAs. (**b**) MFfi and ESfi were measured within ImageJ program and calculated as the mean of the left and right fatty infiltration percentage.

**Figure 4 jcm-14-03109-f004:**
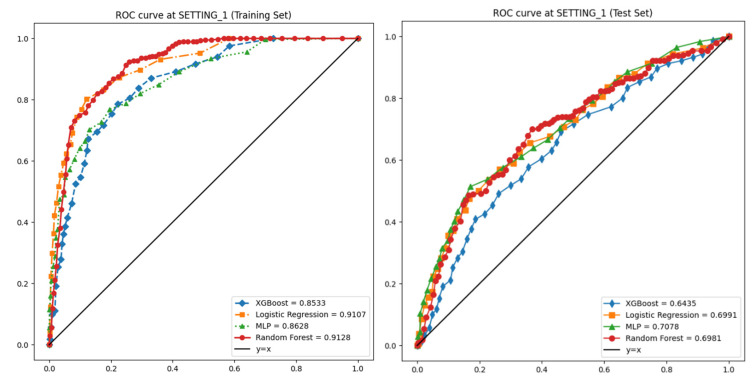
Receiver operating characteristic (ROC) curves for ML models on SETTING_1. **Left**: training set (5-fold CV). **Right**: independent test set.

**Figure 5 jcm-14-03109-f005:**
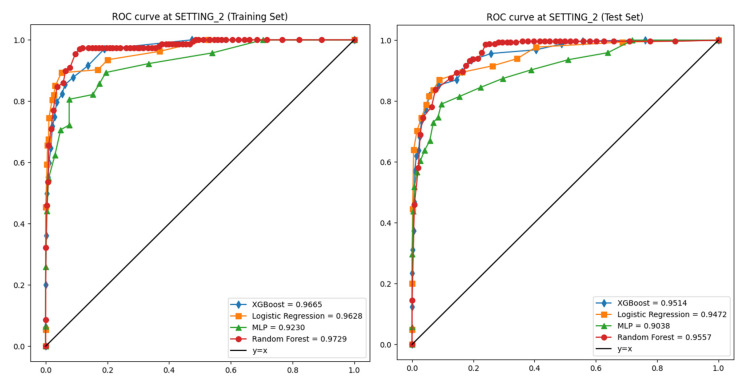
ROC curves for ML models on SETTING_2. **Left**: training set (5-fold CV). **Right**: independent test set.

**Figure 6 jcm-14-03109-f006:**
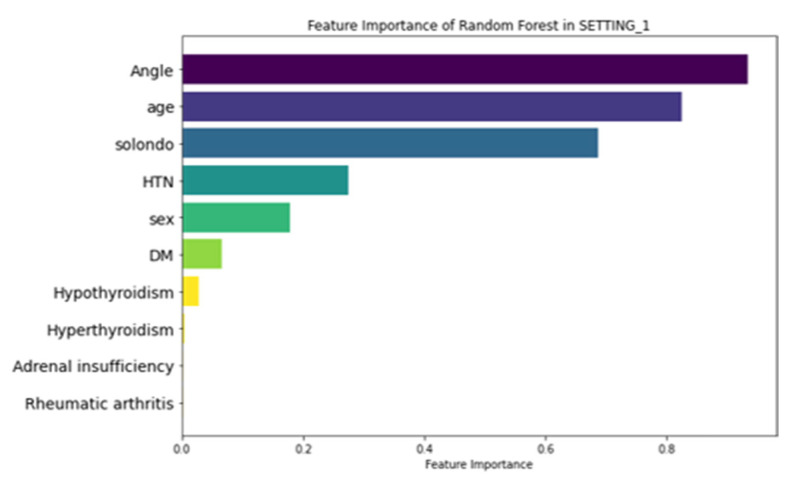
Feature importance of RF in SETTING_1. Angle, local kyphotic angulation of fracture; Solondo, oral steroid use; HTN, hypertension; DM, diabetes mellitus.

**Figure 7 jcm-14-03109-f007:**
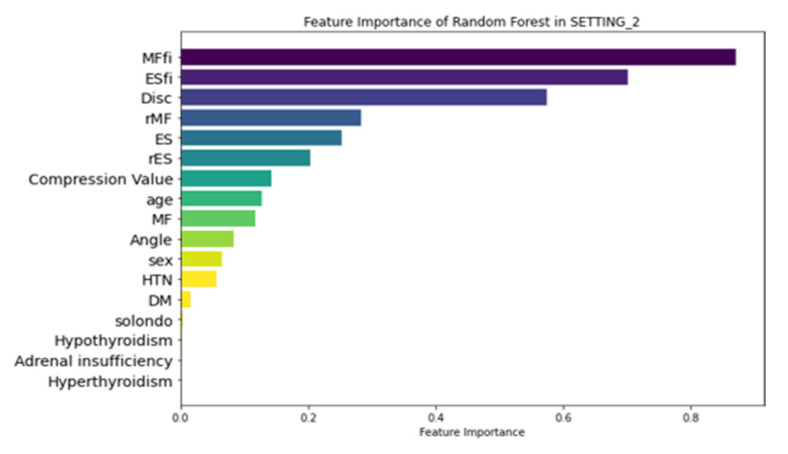
Feature importance of RF in SETTING_2. MFfi, fat infiltration of multifidus; ESfi, fat infiltration of erector spinae; rMF, relative CSA of multifidus; rES, relative CSA of erector spinae; compression value, compression ratio of fracture.

**Figure 8 jcm-14-03109-f008:**
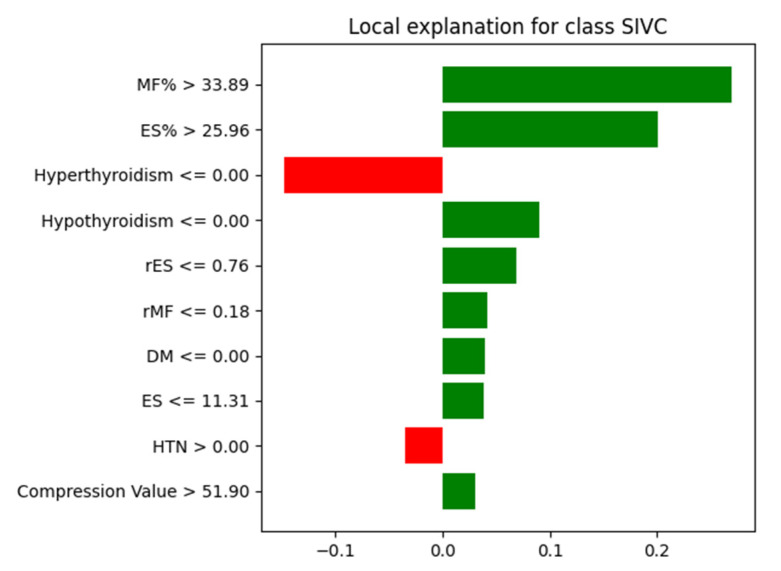
Local explanation for a representative SIVC case using LIME. The top 10 most influential features contributing to the model’s prediction are displayed. Green bars indicate positive contributions to predicting SIVC, while red bars represent suppression. MF% and ES% were the most influential predictors favoring the SIVC class, whereas hyperthyroidism and hypertension slightly decreased the model’s confidence in predicting SIVC.

**Figure 9 jcm-14-03109-f009:**
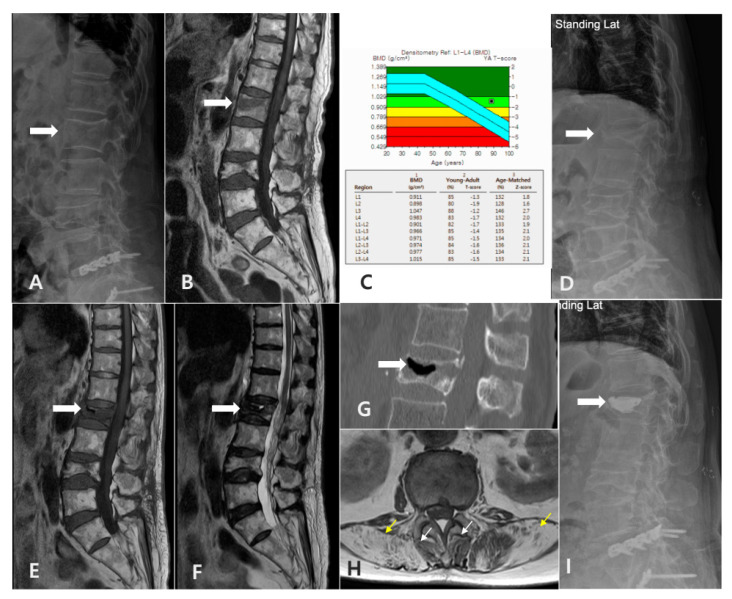
Radiographic findings in an 82-year-old female patient with symptomatic intravertebral vacuum cleft (SIVC). **(A)** Initial lateral radiograph of the lumbar spine, demonstrating mild loss of height at the L2 vertebral body. **(B)** T1-weighted MRI of the lumbar spine showing an **acute** compression fracture of L2. **(C)** Dual-energy X-ray absorptiometry (DXA), demonstrating decreased bone density consistent with **osteopenia**. **(D)** Lateral standing radiograph of the lumbar spine obtained three months after trauma, revealing the further collapse of the L2 vertebral body. **(E,F)** T1- and T2-weighted sagittal MRI sequences of the lumbar spine, illustrating the intravertebral vacuum cleft and associated signal changes. **(G)** CT scan of the lumbar spine depicting a prominent intravertebral vacuum cleft sign. **(H)** T1-weighted axial MRI at the L1–L2 level showing severe fatty infiltration of the paravertebral musculature (white arrows indicate MF, yellow arrows indicate ES). (**I**) This patient eventually underwent cement injection treatment and improved.

**Table 1 jcm-14-03109-t001:** Patient demographics.

Data Source	Characteristics	Number of Patients (%)		Description	*p* Value
		VCF Patients Without IVC	VCF Patients with IVC		
EMR	Age	72.44 ± 1.15	79.85 ± 3.35	Age at diagnosis	<0.05 *
	Sex	F: 464 (71%) M: 190 (29%)	F: 27 (67%) M: 13 (33%)	Sex of patients	0.7746
	DM	P: 76 (12%) N: 578 (88%)	P: 4 (10%) N: 36 (90%)	Diabetes status	0.9549
	HTN	P: 259 (40%) N: 395 (60%)	P: 13 (67%) N: 27 (33%)	Hypertension status	0.4676
	Adrenal insufficiency	P: 7 (1%) N: 647 (99%)	P: 1 (3%) N: 39 (97%)	Adrenal insufficiency status	0.9527
	Hyperthyroidism	P: 1 (1%) N: 653 (99%)	P: 0 (0%) N: 40 (100%)	Hyperthyroidism status	1.0
	Hypothyroidism	P: 10 (2%) N: 644 (98%)	P: 0 (0%) N: 40 (100%)	Hypothyroidism status	0.9169
	Steroid	P: 23 (4%) N: 631 (96%)	P: 1 (3%) N: 39 (97%)	Continuously taking oral steroids = for more than 3 months	1.0
PR (lateral view)	Kyphotic angle (°)	19.89 ± 0.49	25.01 ± 4.21	Kyphotic angulation at VCF	<0.05 *
	Compression Value	34.41 ± 1.27	48.31 ± 6.50	Compression ratio at VCF	<0.05 *
MRI (axial view)	Disc (cm/m^2^)	12.51 ± 0.21	15.14 ± 0.93	CSA of endplate at VCF	<0.05 *
	MF (cm/m^2^)	4.13 ± 0.14	2.95 ± 0.31	CSA of MF at VCF	<0.05 *
	MFfi (%)	16.39 ± 0.53	37.29 ± 4.58	Fatty infiltration percentage of MF	<0.05 *
	rMF	0.34 ± 0.01	0.20 ± 0.02	CSA of relative MF at VCF	<0.05 *
	ES (cm/m^2^)	20.37 ± 0.66	12.12 ± 1.43	CSA of ES at VCF	<0.05 *
	ESfi (%)	10.25 ± 0.51	29.03 ± 4.85	Fatty infiltration percentage of ES	<0.05 *
	rES	1.66 ± 0.05	0.83 ± 0.11	CSA of relative ES at VCF	<0.05 *

VCF, vertebral compression fracture; PR, plain radiographs; MRI, magnetic resonance image; MF, multifidus; ES, erector spinae; CSA, cross-sectional area. *, statistically significant

**Table 2 jcm-14-03109-t002:** Performance of each machine learning model on training set.

Datasets	Models	Accuracy	Specificity	Sensitivity	Precision	F1-Score	AUROC
SETTING_1	Logistic Regression	0.923	0.981	0.57	0.527	0.548	0.911
	Random Forest	0.94	0.977	0.461	0.365	0.407	0.913
	XGBoost	0.935	0.969	0.397	0.296	0.339	0.853
	Multi-Layer Perceptron	0.891	0.991	0.488	0.553	0.519	0.863
SETTING_2	Logistic Regression	0.973	0.992	0.898	0.903	0.901	0.963
	Random Forest	0.99	0.992	0.906	0.93	0.918	0.973
	XGBoost	0.993	0.996	0.917	0.888	0.902	0.967
	Multi-Layer Perceptron	0.972	0.975	0.921	0.892	0.706	0.923

**Table 3 jcm-14-03109-t003:** Performance of each machine learning model on test set.

Datasets	Models	Accuracy	Specificity	Sensitivity	Precision	F1-Score	AUROC
SETTING_1	Logistic Regression	0.757	0.957	0.143	0.527	0.223	0.699
	Random Forest	0.871	0.958	0.185	0.365	0.244	0.698
	XGBoost	0.856	0.949	0.154	0.296	0.192	0.643
	Multi-Layer Perceptron	0.76	0.961	0.151	0.553	0.232	0.708
SETTING_2	Logistic Regression	0.951	0.987	0.606	0.825	0.693	0.947
	Random Forest	0.966	0.982	0.737	0.731	0.731	0.956
	XGBoost	0.962	0.987	0.661	0.796	0.72	0.951
	Multi-Layer Perceptron	0.961	0.978	0.748	0.716	0.723	0.904

**Table 4 jcm-14-03109-t004:** Performance metrics for Random Forest model across 5-fold cross-validation.

Fold	Training Accuracy	Validation Accuracy	Training AUC	Validation AUC
1	0.973	0.968	0.981	0.962
2	0.978	0.959	0.984	0.953
3	0.975	0.964	0.983	0.959
4	0.982	0.971	0.987	0.964
5	0.979	0.967	0.985	0.961
Mean	0.977 ± 0.003	0.966 ± 0.004	0.984 ± 0.002	0.960 ± 0.004

## Data Availability

The patients’ data were collected in Bucheon St. Mary’s Hospital, The Catholic University of Korea. The datasets generated and/or analyzed during the current study are available from the corresponding author (YS) on reasonable request.

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
