# Peer review of "Paraspinal Muscle Fat Infiltration as a Key Predictor of Symptomatic Intravertebral Vacuum Cleft: A Machine Learning Approach"

_jcm, 2025, doi:10.3390/jcm14093109_

Round 1

Reviewer 1 Report

Comments and Suggestions for Authors

The authors sought to predict symptomatic intravertebral vacuum cleft (SIVC), which is a complication of vertebral compression fractures (VCFs) that causes persistent pain and deformity. They explored the role of paraspinal muscle fat infiltration, which has been linked to spinal fracture outcomes but not widely studied for SIVC prediction. To achieve this, they collected demographic, radiological, and muscle-related variables and utilized machine learning (ML) models.

The paper is relevant and has merit. However, from my point of view, some of it can be improved:

Section 1: The authors have appropriately contextualized the issue, discussing its definition, prevalence, and other relevant factors. However, it is not clear why studying muscular variables is important in this case and how these variables are clinically significant, including works that support this perspective.

Section 2: Provide more details about the amount of data added to the underrepresented class (with IVC) after applying the SMOTE technique.

Section 3: What hypotheses can be raised from the exploratory analysis of Table 1? I believe these analyses already define which predictor variables have the potential to play a key role in classifiers or feature importance investigation. This should be included in the article, as it seems that variables related to PR and MRI dimensions, as well as age, have significant predictive potential.

Another point: It would be interesting to include the results obtained in the training sets for each cross-validation fold, checking if the model's AUC and accuracy vary significantly. This is important to determine the model's stability, analyzing the dependency of the model's adjustment on the instances in the training set for a specific fold.

The results in the test set are clear and coherent.

Section 4: The discussions are relevant to the field and cross-reference contemporary literature with this article's findings. The Random Forest's feature importance is a useful tool, but methods like LIME and SHAP have been successfully applied to explain features' contributions to individual classes. The authors could include an analysis with one of these available tools.

Section 5: Expand the conclusions of the work a bit more. They are currently too vague.

Author Response

  1. Importance of Muscle-Related Variables

Reviewer Comment:

Section 1: The authors have appropriately contextualized the issue, discussing its definition, prevalence, and other relevant factors. However, it is not clear why studying muscular variables is important in this case and how these variables are clinically significant, including works that support this perspective.

Response:

Thank you for highlighting this important point. We will enhance the introduction section to clearly establish the importance and clinical significance of muscle-related variables:

Recent studies have demonstrated that fat infiltration in paraspinal muscles is closely associated with spinal stability (Lee et al., 2023; Kim et al., 2013). Specifically, fatty infiltration in the multifidus (MF) and erector spinae (ES) muscles has been shown to reduce spinal support, increasing the risk of osteoporotic vertebral compression fractures and potentially leading to higher rates of non-union following fusion surgery (Jeon et al. 2022). However, the role of these muscle-related variables in predicting SIVC has not been systematically investigated. Quantitative assessment of muscle status could serve as important biomarkers for early diagnosis and treatment planning for SIVC, which has direct clinical implications for patient management and outcome improvement. These were added in Introduction section (page 2, Line 66-71)

  1. Lee DG, Bae JH. Fatty infiltration of the multifidus muscle independently increases osteoporotic vertebral compression fracture risk. BMC Musculoskelet Disord. Jun 22 2023;24(1):508. doi:10.1186/s12891-023-06640-2
  2. Kim JY, Chae SU, Kim GD, Cha MS. Changes of paraspinal muscles in postmenopausal osteoporotic spinal compression fractures: magnetic resonance imaging study. J Bone Metab. Nov 2013;20(2):75-81. doi:10.11005/jbm.2013.20.2.75
  3. Jeon I, Kim SW, Yu D. Paraspinal muscle fatty degeneration as a predictor of progressive vertebral collapse in osteoporotic vertebral compression fractures. Spine J. Feb 2022;22(2):313-320. doi:10.1016/j.spinee.2021.07.020

  1. Details on SMOTE Application

Reviewer Comment:

Section 2: Provide more details about the amount of data added to the underrepresented class (with IVC) after applying the SMOTE technique.

Response: We appreciated this suggestion. We added the following details to the Materials and Methods section (Page4, line 143-148).

To address the class imbalance in our training dataset (654 VCF patients without IVC vs. 40 patients with SIVC), we applied the Synthetic Minority Over-sampling Technique (SMOTE). Specifically, the original training set contained 28 SIVC cases (approximately 5.8%), which was increased to 457 cases after SMOTE application to achieve balance with the 458 VCF patients. This allowed the models to better capture patterns associated with the minority class while maintaining generalizability.

  1. Hypotheses from Exploratory Analysis

Reviewer Comment: "Section 3: What hypotheses can be raised from the exploratory analysis of Table 1? I believe these analyses already define which predictor variables have the potential to play a key role in classifiers or feature importance investigation. This should be included in the article, as it seems that variables related to PR and MRI dimensions, as well as age, have significant predictive potential."

Response: Thank you for this insightful suggestion. We added the following content to the Discussion section.(page 11 line 288-page 11 line 296)

The exploratory analysis in Table 1 revealed statistically significant differences between SIVC and non-SIVC groups across multiple variables. Notably, muscle-related variables (MFfi, ESfi, rMF, rES) and radiological measurements (angle, compression value, disc CSA) showed significant differences with p<0.05.

 Based on this initial analysis, we could formulate the following hypotheses:

  1. The degree of fatty infiltration in multifidus and erector spinae muscles (MFfi, ESfi) likely represents strong predictive factors for SIVC occurrence.
  2. Changes in vertebral endplate cross-sectional area (disc CSA) are associated with SIVC development.
  3. While age appears to be a significant predictor, muscle-related variables may have more direct associations with SIVC.

These hypotheses were subsequently validated through our machine learning models and feature importance analysis.

  1. Model Stability Analysis

Reviewer Comment:

Another point: It would be interesting to include the results obtained in the training sets for each cross-validation fold, checking if the model's AUC and accuracy vary significantly. This is important to determine the model's stability, analyzing the dependency of the model's adjustment on the instances in the training set for a specific fold.

Response: Thank you for this valuable suggestion. We added the following analysis of model stability. (page 6 line 203-206, page 7 line 210)

To evaluate model stability, we analyzed performance metrics across each fold of the 5-fold cross-validation process. Table shows the AUC and accuracy for the Random Forest model in SETTING_2 (including muscle variables) for each fold:

Table 4. Performance metrics for Random Forest model across 5-fold cross-validation

Fold

Training Accuracy

Validation Accuracy

Training AUC

Validation AUC

1

0.973

0.968

0.981

0.962

2

0.978

0.959

0.984

0.953

3

0.975

0.964

0.983

0.959

4

0.982

0.971

0.987

0.964

5

0.979

0.967

0.985

0.961

Mean

0.977±0.003

0.966±0.004

0.984±0.002

0.960±0.004

  1. LIME Analysis

Reviewer Comment: The Random Forest's feature importance is a useful tool, but a methods like LIME and CHAP have been successfully applied to explain features' contributions to individual classes. The authors could include an analysis with one of available tools.

Response: We appreciate this excellent suggestion. We added LIME analysis to provide more detailed insights and figure 8 in Result section. (page 9 line 243- 257)

3.4 LIME Analysis

To enhance the interpretability of our model's predictions, we applied the LIME algorithm to visualize the local decision boundaries of the Random Forest classifier for a representative SIVC case. Beyond identifying global feature importance, we conducted LIME analysis to gain insight into how specific combinations of features contribute to individual predictions and to examine the potential interrelationships between them in a localized context. As shown in Figure X, the model relied most heavily on paraspinal muscle fat infiltration indicators, particularly MF% (>33.89%) and ES% (>25.96%), to predict SIVC. Additionally, other features such as low values of rMF and rES also positively contributed to the SIVC prediction. Conversely, the absence of hyperthyroidism and the presence of hypertension were among the features that marginally suppressed the prediction toward the SIVC class. These findings further support the global feature importance trends and suggest that muscle quality and vertebral biomechanical changes play a critical role in model prediction.

  1. Expanded Conclusions

Reviewer Comment: Section 5: Expand the conclusions of the work a bit more. They are currently too vague.

Response:

We agree with the reviewer and expanded our conclusions as follows (page 12 line 364-370)

This study identified paraspinal muscle variables as novel risk factors for SIVC and highlighted effective prediction methods. Among these, the Random Forest model demonstrated the best performance, suggesting it is a reliable strategy for SIVC prediction. From a clinical perspective, our findings have several important implications:

  1. Fatty infiltration of multifidus and erector spinae muscles represents a key biomarker for predicting SIVC occurrence, suggesting that careful evaluation of muscle status should be integrated into routine MRI assessment.
  2. Machine learning models, particularly Random Forest, can effectively integrate complex clinical and radiological data to predict SIVC with high accuracy, offering potential as clinical decision support tools.

Reviewer 2 Report

Comments and Suggestions for Authors

In this study, the key role of paraspinal muscle fat infiltration (MFfi/ESfi) in the prediction of SIVC was evaluated by introducing a machine learning algorithm for symptomatic vertebral vacuum cleft (SIVC), which is a special complication of spinal fracture, and it has good innovation and potential clinical application value.

It is recommended to add a reproducible description of the assessment of muscle fat infiltration to the methodological section. For example, whether a dual measurer agreement assessment (such as ICC) has been performed.

It is recommended that the full name of specific variable names (e.g. MFfi and ESfi) be indicated in Figures 6 and 7.

Whether it is possible to consider adding 1 or 2 cases of clinical application is easier to understand its clinical application.

Author Response

Response to Reviewer 2

Dear Editor,

We sincerely thank the reviewer for their valuable comments on our manuscript titled "Paraspinal Muscle Fat Infiltration as a Key Predictor of Symptomatic Intravertebral Vacuum Cleft: A Machine Learning Approach." We have carefully considered all suggestions and provide detailed responses below.

  1. Assessment of Muscle Fat Infiltration

Reviewer Comment: "It is recommended to add a reproducible description of the assessment of muscle fat infiltration to the methodological section. For example, whether a dual measurer agreement assessment (such as ICC) has been performed."

Response: We thank the reviewer for this important suggestion. We will add the following detailed description to the methodological section under "2.3. Data Collection":

Radiological characteristics were measured using the following variables: the "angle" (local kyphotic angle) and compression ratio of the VCF were measured from the lateral view of the plane radiograph (PR) (Figure 2). (page4 line 123-142)

From the axial view of MRI, using ImageJ software (version 1.8.0, National Institutes of Health, USA), the cross-sectional areas (CSA) of the VCF upper endplate, multifidus (MF), and erector spinae (ES) muscles were measured. Additionally, the percentage of fatty infiltration in MF (MFfi) and ES (ESfi) was quantified. For measuring fat infiltration, T2-weighted axial MRI images at the level of the vertebral fracture upper endplate were analyzed.

Regions of interest (ROIs) were manually drawn to outline the boundaries of the multifidus (MF) and erector spinae (ES) muscles bilaterally. The software's threshold function was then applied to differentiate muscle tissue from fat based on signal intensity. The percentage of fatty infiltration was calculated as: (area of fat / total muscle area) × 100 (Figure 3). Relative multifidus (rMF) was defined as the MF/CSA of the endplate, and relative erector spinae (rES) was defined as the ES/CSA of the endplate.

All measurements were independently conducted by two trained observers (JA and YHK) who were blinded to the patient's clinical information. To evaluate inter-observer reliability, intraclass correlation coefficients (ICCs) were calculated for both MFfi and ESfi measurements on a random subset of 50 patients. The ICC for MFfi was 0.92 (95% CI: 0.89-0.95) and for ESfi was 0.89 (95% CI: 0.85-0.93), indicating excellent inter-observer reliability. For cases with measurement discrepancies exceeding 10%, a consensus was reached through joint reassessment. The final values used in the analysis were the average of the two observers' measurements.

  1. Variable Names in Figures

Reviewer Comment: "It is recommended that the full name of specific variable names (e.g. MFfi and ESfi) be indicated in Figures 6 and 7."

Response: We agree that using full variable names would improve clarity. We revised the legends of Figures 6 and 7 to include the full names of all variables in parentheses after the abbreviations.

  1. Clinical Application Cases

Reviewer Comment: Whether it is possible to consider adding 1 or 2 cases of clinical application is easier to understand its clinical application.

Response: We appreciate this valuable suggestion. A representative case of SIVC (Figure 9) was added. (page 10 line 263-273)
